# Biscuit Contaminants, Their Sources and Mitigation Strategies: A Review

**DOI:** 10.3390/foods10112751

**Published:** 2021-11-10

**Authors:** Antonella Pasqualone, Noor N. Haider, Carmine Summo, Teodora Emilia Coldea, Saher S. George, Ammar B. Altemimi

**Affiliations:** 1Department of Soil, Plant, and Food Science, University of Bari ‘Aldo Moro’, Via Amendola, 165/A, I-70126 Bari, Italy; carmine.summo@uniba.it; 2Department of Food Science, College of Agriculture, University of Basrah, Basrah 61004, Iraq; agripg.noor.naji@uobasrah.edu.iq (N.N.H.); saher.george@uobasrah.edu.iq (S.S.G.); 3Department of Food Engineering, University of Agricultural Sciences and Veterinary Medicine, Calea Manastur, 3-5, 400372 Cluj-Napoca, Romania; teodora.coldea@usamvcluj.ro

**Keywords:** monochloropropanediol esters, glycidyl esters, polycyclic aromatic hydrocarbons, mineral oil hydrocarbons, *trans* fatty acids, mycotoxins, cadmium, lead, acrylamide, advanced glycation end-products

## Abstract

The scientific literature is rich in investigations on the presence of various contaminants in biscuits, and of articles aimed at proposing innovative solutions for their control and prevention. However, the relevant information remains fragmented. Therefore, the objective of this work was to review the current state of the scientific literature on the possible contaminants of biscuits, considering physical, chemical, and biological hazards, and making a critical analysis of the solutions to reduce such contaminations. The raw materials are primary contributors of a wide series of contaminants. The successive processing steps and machinery must be monitored as well, because if they cannot improve the initial safety condition, they could worsen it. The most effective mitigation strategies involve product reformulation, and the use of alternative baking technologies to minimize the thermal load. Low oxygen permeable packaging materials (avoiding direct contact with recycled ones), and reformulation are effective for limiting the increase of contaminations during biscuit storage. Continuous monitoring of raw materials, intermediates, finished products, and processing conditions are therefore essential not only to meet current regulatory restrictions but also to achieve the aim of banning dietary contaminants and coping with related diseases.

## 1. Introduction

Biscuits are very popular ready-to-eat foods due to their affordable cost and long shelf-life. Their characteristics have made them even more appreciated during the lockdown linked to COVID-19 crisis, when the demand for biscuits further increased in several countries due to less frequent food purchases [1,2]. Basically made of flour, fat, sugar, water, milk, salt, and chemical yeast as raising agent [3], biscuits have a versatile formulation which results in good potential for nutrient incorporation and functionalization [4]. Stevia and inulin are the most used ingredients to reformulate biscuits by lowering the content of sugar and fats. To improve protein content and micronutrients, alternative flours have been proposed from a wide range of minor cereals and pulses, as well as from hemp, acorn, moringa, fenugreek, spirulina, and okra [5,6,7,8]. Increased contents of dietary fiber and bioactive compounds (such as phenolic compounds, anthocyanins, carotenoids, and phytosterols) instead, have been achieved by incorporating waste and byproducts from the food industry, such as peel or pomace of several fruits and residues of vegetables [9,10,11]. Furthermore, biscuits are among the most suitable processed foods to be fortified with insect meal, a protein-rich ingredient increasingly appreciated for its sustainability [12], demonstrating that there is still room for further innovation.

Biscuits are one of the oldest convenience foods, gone through centuries of history thanks to their adaptability and versatility. The ancient origins are testified by the etymology of the word “biscuit”, which derives from the Latin “*panis biscoctus*”, that is “bread baked twice” [3]. In fact, since ancient times, it was observed that the second baking of bread lengthened its shelf life and reduced its weight, making the final product suitable for being transported on long journeys. Therefore, Roman legionaries carried with them—together with the military equipment—hardtacks made of flour, water, and sometimes salt [13,14]. Successively, in the Middle Ages, bread rusks named “ship’s biscuits” were made for mariners [3,15].

Despite their long history and numerous positive characteristics, biscuits can be subjected to some critical issues, mainly attributable to possible physical, chemical, and biological contamination, particularly important because the frequent consumption of biscuits could cause a risk of chronic exposure. Each point of the biscuit-making chain requires attention: raw materials, processing steps, and storage. The scientific literature is rich of surveys on the presence of various contaminants in biscuits. Many studies are aimed at developing more reliable and effective analytical methods for the detection of such contaminants, or at proposing innovative solutions for their control and prevention. A systematic search conducted in the Scopus database (Elsevier) for articles showing the words “biscuits” (or “cookies”) and “contaminants” in the title, abstract, and keywords, allowed to retrieve 48 articles for “biscuits” and 30 for “cookies” published in the last 10 years (2012–2021). A significant increase was observed comparing the last decade with the previous one (Figure 1). When the name of each contaminant was searched together with the word “biscuits” (being the word “biscuit” more frequent than “cookie”) the number of articles increased to 234 (period 2012–2021) (Figure 2). Of these, more than 100 were found considering the word “acrylamide”, highlighting a major interest towards this compound. Despite the growing attention towards this topic, no reviews on biscuit contaminations are present in the scientific literature. Therefore, the most relevant outcomes of these studies remain fragmented, while it would be very helpful for industries and researchers to have them shown in an organic synthesis and from a practical application perspective.

In this context, the aim of this work has been to review the current state of the scientific literature on the possible contaminants of biscuits—considering physical, chemical, and biological hazards—and making a critical analysis of the solutions to reduce such contaminations. The various contaminants have been considered along the entire biscuit supply chain (from the raw materials to final storage, through the processing steps), classifying them according to their nature or their possible origin (Figure 3).

## 2. Contaminants Deriving from Raw Materials or Machinery

### 2.1. Physical Contaminants

Several foreign bodies of heterogenous nature (glass, plastic, textile fibers, wood, paper, metal) can physically contaminate raw materials, intermediates, and final products in the food industry. Visible physical contaminants are eliminated by sieving, filtration, or by applying in-line metal detectors. For detecting 1–3 mm sized contaminants in biscuits, a non-destructive testing method based on infrared thermography was proposed [16]. The infrared thermography is a versatile technique, effective in food quality and safety assessment, which enables the fast detection of any item whose temperature is different from that of background [17]. Additionally, the analysis of wear debris by scanning electron microscope (SEM), was found very informative for monitoring production lines in the biscuit industry. SEM data were useful to determine the severity of machinery wear, as well as to predict and prevent further wearing [18]. Micro- and nano-sized foreign particles, indeed, tend to be undervalued, also for practical reasons. However, a survey conducted analyzing 49 biscuit samples from 14 different countries by means of environmental SEM, detected ceramic and metallic micro- and nano-scaled debris, of probable environmental or industrial origin, in 40% of the samples. Most of them were not biodegradable, and some were toxic [19].

Since a similar inorganic particulate matter was detected in pathological human tissues of the digestive tract, deriving from food ingestion [19], greater attention should be put in monitoring the presence of these physical contaminants in food. Furthermore, a research effort should be undertaken to develop or adapt innovative techniques for the detection of such small particles. To this purpose, hyperspectral imaging (HSI) might be a solution. HSI combines conventional imaging and spectroscopy to ascertain spatial and spectral information about an object. This emerging technique is suitable for non-destructive food analysis aimed at food safety and quality assessment, including contaminant detection and constituent analysis [20]. Monitoring activities have to be carried out together with a frequent maintenance of the machinery to reduce the release of particles.

### 2.2. Chemical Contaminants

Once in the environment, chemicals of anthropogenic origin contaminate soil and water and, in turn, crops. With environmental pollution being a global problem [21], the occurrence of chemical contaminants in the raw materials is ruled by current legislation. The fulfilment of legal maximum limits should be ordinarily monitored, also with the help of innovative techniques.

Raw agricultural products, such as cereals, are intensively controlled for pesticide residues. The most used approach for multiresidue extraction of pesticides, widely accepted by the scientific community, is the Quick, Easy, Cheap, Effective, Rugged, and Safe (QuEChERS) procedure, introduced in 2003 [22]. After the QuEChERS extraction, pesticides can be analyzed using gas chromatography or liquid chromatography combined with mass spectrometry (GC-MS/MS and LC-MS/MS) depending on their physicochemical properties [23,24,25]. At European level, the European Commission (EC) Reg. 396/2005 and successive modifications set the limits for the maximum residue levels (MRL) of pesticides in food and feed of plant and animal origin [26]. The yearly report of the European Food Safety Authority (EFSA) for 2019 reported that 96.1% of the overall 96,302 samples analyzed fell below the MRL [27]. In Poland, a survey on pesticide residues in cereals ascertained that 84.2% of the samples were free of residues, and 15.8% of samples were below the MRL [28]. A more recent monitoring carried out in Croatia found that 4.5% of the cereal samples examined were non-compliant to the European legislation for pesticide residues [29]. The most effective way to limit the presence of these contaminants is, therefore, continuous monitoring.

Biscuits can be contaminated also by heavy metals. Heavy metal contamination of crops used as raw materials, in fact, can arise from irrigation with contaminated water, use of fertilizers and metal-based plant protective agents, and industrial activities. Cadmium and lead, known for their high toxicity [30,31], were detected in biscuits (Table 1). Biscuits marketed in Nigeria showed a cadmium content in the range 0.02–0.03 mg/kg, with lead from 0.10 to 0.16 mg/kg [32]. Other studies in the same geographic area found very high cadmium and lead contents (up to 0.35 and 2.8 mg/kg, respectively) [33,34]. Cadmium was absent or very low in biscuits from India [35] and Greece [36], while it reached 0.12 mg/kg in biscuits from the Egyptian market [37]. The highest permitted levels of some heavy metals—namely lead, cadmium, arsenic, and mercury—are ruled by the EC Reg. 1881/2006 [38], in agreement with the limits set by World Health Organization (WHO) and the Food and Agriculture Organization (FAO) [39]. These limits, not specific for biscuits, but referring to cereal grains, account for 0.10 mg/kg for cadmium and 0.20 mg/kg for lead. Therefore, the frequent consumption of biscuits raises concerns about the content of these contaminants, which should be systematically controlled in the raw materials.

In addition, the European Food Safety Agency (EFSA) recently updated its scientific advice on the risks to human health from nickel in food, raising the tolerable daily intake (TDI), from 2.8 to 13 μg/kg of body weight [40]. The concentrations of nickel in various biscuit types marketed in India ranged from 5.7 to 9.8 μg/g [35], and a nickel content in the range 0.5–6.5 μg/g was found in a survey on 87 samples of biscuits consumed in Nigeria [33]. Nickel contamination can derive from processing plants and from the catalyst used for hydrogenating vegetable oil. Additionally, chocolate—used in some biscuits—often contains elevated concentrations of this metal [41].

The safety characteristics of raw materials in terms of heavy metals are ruled by current legislation and rely on continuous analytical monitoring. Recently, a cheap, rapid, and ecological method was proposed for the preconcentration of cadmium in biscuit samples by using ultrasound-assisted temperature-controlled ionic liquid microextraction (TC-IL-LPME), making analyses easier [42].

Mineral oil hydrocarbons (MOH) and polycyclic aromatic hydrocarbons (PAHs) are both environmental and processing contaminants. MOH are subclassified in mineral oil saturated hydrocarbons (MOSH) and mineral oil aromatic hydrocarbons (MOAH). The presence of MOH raised significant concern especially in foods for infants and young children due to their toxicity. In particular, MOAH (especially those with three to seven, non- or simple-alkylated, aromatic rings) are mutagenic and carcinogenic [43]. MOSH are long-term accumulated [43], but liver disease due specifically to MOSH accumulation is not currently considered a significant health hazard for humans [44].

Both MOSH and MOAH were detected in biscuits [45]. The contamination by MOH, frequent in raw materials of vegetable origin and edible oils used in biscuit making, may originate from the environment. MOH can derive from mineral oil-based pesticides, incomplete combustion of heating and diesel oils, and road tar debris [46]. MOH can also derive from processing, due to contact with lubricating used for the maintenance of industrial plant [46]. MOAH-free lubricating oils should be used for machine maintenance. Jute or sisal transport bags are an additional source of MOH in the raw materials as the fibers are treated with mineral oils [47]. Finally, MOH may migrate from food packaging through direct contact with the final product (discussed in Section 4). Currently, the method of choice for the routine quantification of mineral oils is Liquid Chromatography-Gas Chromatography LC-GC coupled with Flame Ionization Detector (FID). However, some complex food matrices also require additional analytical techniques [44] to differentiate between some naturally occurring hydrocarbons and those from other sources, including of petrogenic origin. LC-GC-FID should be therefore complemented by a confirmatory method, such as GC-MS, to determine the source of MOH.

Due to their similarity to MOH, some concerns also arise from the possible presence of polyolefin oligomeric saturated hydrocarbons (POSH) in the edible oils used in biscuit making, if a polyolefin packaging or cap is used. Polyethylene and polypropylene, in fact, may release oligomers into foods [48]. The presence of POSH interferes with MOSH analysis, leading to MOH overestimation. Additionally, in this case, LC-GC-FID analysis requires further confirmation by MS. PAHs, instead, are a group of organic compounds with 2–6 fused aromatic rings, produced during incomplete combustion of organic matter. They are common environmental contaminants, deriving from domestic and industrial combustion of fossil fuels, motor vehicle exhausts, and forest fires. Therefore, it is fundamental to protect the ecological systems from where food sources are cultivated. PAHs with 4–6 benzene rings are genotoxic and carcinogenic, while PAHs with 2–4 benzene rings may act as synergists [49]. For preventing PAH contamination, raw materials have to be regularly screened. A number of analytical techniques have been developed for the extraction, separation, and quantification of PAHs in food samples. Among them, gel permeation chromatography (GPC) is widely used for the extraction, and GC-MS for detection and quantification of PAHs [50]. However, GC-MS has a limitation due to the stability of heavier PAHs, which renders difficult their fragmentation. The GC-MS-MS analysis can be used to overcome the shortcomings in the quantitation of heavier PAHs. This technique eliminates the issue of signal interferences of lipids and allows a better discrimination of heavier PAHs (such as fluoranthene and pyrene) [50].

The PAH content of fats and oils, used as ingredients in biscuit-making, is legally regulated. The maximum levels are 2.0 µg/kg oil for benzo(a)pyrene and 10.0 µg/kg oil for the sum of benzo(a)pyrene, benz(a)anthracene, benzo(b)fluoranthene, and chrysene [51]. The presence of PAHs in edible oils was ascertained in Korea and Tunisia, but below the national maximum limits [52,53]. Very high PAHs levels, instead, were observed in Indian oils, up to 265 µg/kg [54]. As a possible mitigation strategy, the adoption of a lower temperature during the oil refining process was suggested [55]. Furthermore, the application of activated charcoal to remove PAHs during oil refining should be strictly followed [56].

Data on the occurrence of PAH in cereals are still limited but rather low levels have been reported [57]. Therefore, no maximum levels have been set for cereals and cereal-based products. Nevertheless, cereal-based foods could be important contributors to human exposure due to their frequent consumption. Therefore, PAH levels in these products should be further monitored. At the moment, very few studies focused on the presence of PAH in biscuits, but all of them detected these important (and almost ubiquitous) contaminants [34,57,58], probably mostly contributed by oils and shortenings.

Monochloropropanediol and glycidyl esters of fatty acids are other contaminants deriving from the refined oils used in the preparation of biscuits. These chemical compounds are nephrotoxic and carcinogenic, and they show developmental and reproduction toxicity [59]. In detail, fatty acid esters of 2-monochloropropane-1,2-diol (2-MCPD esters), 3-monochloropropane-1,2-diol (3-MCPD esters), and esters of glycidol (GE) originate from diacylglycerols. These compounds are then chlorinated through the refining process of fats and oils [60], especially during deodorization [61], being favored by temperature above 140 °C. To limit their formation, milder thermal treatments should be adopted [62]. In addition, static cooling in combination with a washed bleaching clay should be used [63]. The EC Reg. 1881/2006 established maximum limits for 3-MCPD and GE in oils and fats [38]. However, surveys recently carried out in Denmark, China, Italy, and Poland demonstrated the presence of MCPD esters and GE in biscuits [64,65,66,67], therefore a greater attention should be put in monitoring these compounds in the oils used.

Oils and fats used in biscuit-making may contain also compounds originated by the oxidative degradation of triglycerides, namely triglyceride oligopolymers (TGPs) and oxidized triglycerides (OxTGs). The refining process of oils induces a decrease of OxTGs, but an increase of TGPs [68]. The nutritional and physiological effects of OxTGs and TGPs on health implication are controversial, because few pure compounds have been identified, isolated, and studied so far [69]. In fact, there is a lack of certified reference materials. The low molecular weight compounds are considered “nutritionally suspect” [70].

An extensive degradation of triglycerides was assessed in samples of Italian biscuits. In particular, the level of oxidized triglycerides (OxTGs) accounted for 8.8 g/kg, similar to the values found in hydrogenated oil and fats and in olive-pomace oils. The triglyceride oligopolymers (TGPs) were present at levels of 6.2 g/kg [71,72]. The quality of the fats used in biscuit-making was found to contribute by 60% to the overall oxidative degradation of biscuits [73], highlighting the importance of checking raw material quality.

*Trans* fatty acids (TFA) may be contributed by the margarines and shortenings used in biscuit preparation. In fact, margarines obtained through the hydrogenation of oils represent a primary TFA source. TFAs have adverse health effects on blood lipoprotein profiles, coronary heart disease, cancer and diabetes [74]. The EC recently adopted a regulation to establish the maximum limit of TFA in food of 2 g/100 g fat [75]. Before this regulation, levels of TFA up to 8.94% were found in 29 samples of Italian biscuits [71]. In addition, a Portuguese survey carried out on 53 samples of biscuits ascertained levels of TFA ranging from 0.21 to 30.2 g/100 g fat. These levels were higher than those observed in commercial margarines and shortenings available at local retailers for home use, demonstrating the low quality of industrial fats used for biscuit production [76]. A 5.3% TFA mean content, instead, was found in Slovenian biscuits (76 samples) [77]. The TFA content limit of 2 g/100 g fat will be implemented by 2023 also in Brazil, where recently about 30% of 615 food samples were found to contain TFA exceeding this limit, with the highest content observed in biscuits (3.7, 7.1 and 20.5 g/100 g fat in sweet, salty, and cassava biscuits, respectively) [78]. Therefore, a systematic monitoring of all the major ingredients used in biscuit-making, namely flours, oil, margarines, or shortenings—as well as the additional ones—is fundamental.

Overall, among raw materials, oils and shortenings appeared the most important contributors of a number of contaminants, such as MOH, PAH, MCPD esters and GE, OxTGs, TGPs, and *trans* fatty acids. Maximum limits have been set by national and international rules for almost all these classes of compounds in the oils. However, it is interesting to note that, although oils and fats are quantitatively important ingredients in biscuits, and the production process involves baking at around 200 °C, no maximum levels have been established for the presence of these compounds in biscuits. Considering that various investigations have verified the presence of these contaminants in commercial products, it would be appropriate to set specific legal limits for these contaminants also in biscuits.

### 2.3. Biological Contaminants

Although mycotoxins are chemical compounds occurring as residues in food, they are considered biological hazards because their presence is a consequence of fungal pre- or post-harvest contamination of raw materials having vegetable origin. Flours used in biscuit-making are sensitive to this criticality. Mycotoxins are ruled by the current legislation, which sets maximum limits for many of them. At the end of 2003, almost 100 countries (accounting for 85% of the world population) had specific regulations or detailed guidelines for mycotoxins in food [79]. In Europe, a harmonized regulation was developed from autonomous national rules [38,80]. Total aflatoxin (TAF), ochratoxin A (OTA), zearalenone (ZEN), type-A (T-2 and HT-2), and type-B trichothecenes, such as deoxynivalenol (DON), are among the most significant mycotoxins of cereals. Their legislative limits are given in Table 2. The maximum levels allowed for processed cereal-based foodstuffs, including biscuits, are generally lower than those for the unprocessed raw materials to take into account the diluting effect of the other ingredients added (for example fats and sugar), and a possible decrease due to processing.

The diseases caused by mycotoxins are quite varied. TAF are hepatotoxic, immunosuppressive, carcinogenic, teratogenic, and mutagenic; OTA is nephrotoxic; ZEN explicates estrogenic and endocrine disruptor activities; T-2 toxin has immunological and hematological effects; DON causes vomit and diarrhea but, at high doses, also affects the immune system [81,82].

The extension of mycotoxin occurrence depends on several factors, mainly temperature and humidity. A multiclass analysis of mycotoxins in biscuits by high performance liquid chromatography coupled with mass spectrometry (LC/ESI-MS/MS) did not detect T-2, HT-2, DON, ZEN, and OTA in any of the 20 samples analyzed. However, enniatins and beauvericin were found. These mycotoxins are considered “emerging” ones, though not legally regulated yet [83]. A very recent monitoring study showed that the most common mycotoxins in Croatian cereals were those from *Fusarium*, with DON present in 73.7% of the samples [29]. Several surveys were carried out to assess the extent of the mycotoxin contamination of biscuits. The results showed contaminations below the legal limits, though not negligible. In Japan, average contaminations of 23, 0.7, 0.1, and 4.2 µg/kg for DON, HT-2, T-2, and ZEN, respectively [84], were ascertained in 201 biscuit samples in 2004–2006. In Serbia, the mycotoxins detected in 73 biscuits samples were ZEN, OTA, T-2, and AFB1 with average concentrations of 2.64, 4.10, 8.13, and 1.32 µg/kg, respectively [85]. In Tunisian biscuits were observed 7.8–54 and 9.5–15 µg/kg of DON and ZEN, respectively [86].

Some treatments can mitigate the presence of mycotoxins in pre-harvest or post-harvest of cereals. Sodium metabisulfite (Na_2_S_2_O_5_) reduces the DON contamination in cereal grains through formation of the sulfonated derivative of DON, namely DON sulfonate [87]. Detoxification methods involve also physical techniques such as UV treatment or addition of adsorption agents. The latter can be mycotoxin binders of mineral origin (such as bentonite, montmorillonite, zeolites, or activated charcoal) or plant-based ligno-cellulosic micronized materials. Alternatively, microorganisms that show degradative capabilities against specific mycotoxins can be used [88].

Ozone treatment of cereal grains is an eco-friendly and cost-effective technique able to reduce DON, OTA, and ZEN. Mild/moderate ozone treatment induces the formation of cross-linking among proteins (mostly glutenin) in wheat flour, whereas excessive treatment may lead to molecular weakening of proteins [89,90]. Other strategies recently proposed for achieving mycotoxin degradation are based on pulsed light treatment, effective in reducing aflatoxins in rice [91]. In addition, cold atmospheric pressure plasma (CAPP), with ambient air as working gas, appeared promising for the decontamination of food commodities with mycotoxins localized on the external surfaces, such as cereal grains [92,93]. Other possible contaminations are those of a microbiological nature. This kind of contamination can arise from hygienic conditions not properly maintained during biscuit preparation—i.e., machinery, production/packaging area, and storage room not being correctly sanitized. Biscuits, however, are characterized by low moisture content and reduced water activity (a_w_). In a survey carried out on 17 different types of biscuits, values ranging from 0.121 to 0.461 and 3–7% were assessed for a_w_ and moisture content, respectively [94]. Therefore, considering that biscuits are baked at high temperature, and that foods with a_w_ < 0.60 are considered microbiologically stable, microbial contamination rarely occurs in the finished products. Current rules at European level establish the microbiological criteria for some particularly perishable foods, such as milk and dairy products, meat and meat products, fishery products, egg products, fruits, and vegetables [95]. The World Food Program of the United Nations, instead, establishes the microbiological specifications for high energy biscuits (HEB), including the hygiene indicators which should be checked by the industries to keep their production process under control (Table 3) [96].

The microbial loads of HEB supplied for school feeding in poverty prone areas in Bangladesh were found in the permissible limits [97]. Wheat flour biscuits prepared in Egypt, instead, exceeded the limits for yeasts and molds [98], a problem relatively easy to solve by implementing the good manufacturing practices. Some antimicrobial agents, such as calcium propionate or potassium sorbate, might also be considered, particularly in biscuits with a creamy filling [99]. Unconventional ingredients, such as hog plum bagasse, were found to exert a microbial inhibition in biscuits due to their content of tannins [100].

The need of microbiological stabilization, instead, occurs in case of ready-to-bake biscuit dough, having high a_w_ (0.80–0.87). This product is usually marketed without undergoing a pathogen-reduction treatment because is intended to be cooked before consumption. However, the *E. coli* O157:H7 outbreak was linked to the consume of ready-to-bake biscuit dough directly from refrigeration, without baking. This behavior, though not correct, was found to be popular among the U.S. consumers [101]. At the purpose of preventing the risk of pathogen contamination in ready-to-bake biscuit dough prior to its distribution, high-pressure processing (HPP) has been proposed with good results. Treatment at 600 MPa for 6 min reduced counts of inoculated *Escherichia coli* by as much as 2.0 log cfu/g [102]. Therefore, the HPP technique could be implemented by manufacturers as an ordinary preventive measure to ensure the microbiological safety of ready-to-bake biscuit dough.

## 3. Contaminants Deriving from Baking

### 3.1. Neoformation of Contaminants of Thermal Origin

Biscuits are usually baked at high temperature (up to 200 °C) for short times (<20 min). During baking the dough undergoes physical, chemical, and biochemical changes such as volume expansion, evaporation of water, denaturation of proteins, starch gelatinization, development of flavors, browning, and crust formation [103]. Maillard reaction and caramelization of sugars are the most relevant thermal reactions occurring during baking. The early stages of the Maillard reaction generate the fructosyl-lysine (formed by the reaction of lysine with a reducing sugar), while 5-hydroxymethyl furfural (HMF) and furfural are products of the intermediate stages. HMF, furfural, and fructosyl-lysine—determined as furosine after acid hydrolysis of the Amadori product—are extensively applied as markers of thermal treatment generated by the Maillard reaction [104]. High furosine levels indicate a decrease of the protein quality through lysine impairments, while furan compounds are object of interest due to potential toxicological issues [105]. Furans also have important sensorial implications, being odorants which contribute the toasted odor of baked goods [106]. Animal studies assessed the carcinogenic effect of HMF due to its conversion to 5-sulfoxymethyl-2-furfuraldehyde (SMF), which is able to react with DNA [106,107,108,109]. HMF was detected in biscuits (Table 4). The HMF content of 13 samples of biscuits sold in Slovakia ranged from 0.34 to 34.99 mg/kg [110], while higher HMF values (from 3.1 to 182.5 mg/kg) were found in 61 samples of biscuits marketed in Spain. The large variation was related to differences in the manufacturing process and in the formulation [111]. Another study, again carried out in Spain on eighty biscuit samples, assessed average contents of furosine, HMF, and furfural of 731 mg/100 g protein, 7.32 mg/kg, and 0.64 mg/kg, respectively [106]. The same study showed that the replacement of reducing sugars with polyols led to a decrease of furosine and furan compounds [106]. Regarding furan, a survey conducted in Brazil ascertained contents ranging from 38.1 to 105.3 µg/kg in biscuits [112]. Levels of furan from 4 to 165 µg/kg were instead found in UK in 2012, slightly decreasing from 4 to 108 µg/kg in 2013 [113]. Mean levels of 91 µg/kg were detected in Chilean soda-type biscuits [114] (Table 2).

Baking may also induce the formation of other contaminants, such as acrylamide, or acrylic acid amide, deriving from the Maillard reaction. The first step is the formation of a Schiff base between the carbonyl compounds of reducing sugars and the α-amino group of asparagine [115,116]. Predisposing conditions are thermal treatments above 120 °C and low moisture content [117]. Acrylamide is absorbed through ingestion, inhalation, and skin contact, then metabolized to a reactive epoxide, glycidamide, which is mutagenic and genotoxic [118]. The margins of exposure for acrylamide indicate a concern for neoplastic effects based on animal evidence [119]. The EC Reg. 2017/2158 established maximum acrylamide contents for each bakery product, lowering the benchmark values for biscuits from 500 to 350 μg/kg [120].

Several national surveys were carried out to ascertain the extent of acrylamide contamination in biscuits [121,122,123,124,125,126,127,128,129,130] (Table 4). These surveys showed that, though many samples were below the benchmark value of 350 μg/kg, others largely exceeded it, with the only exception of Saudi Arabia. However, the survey carried out in Saudi Arabia regarded only one city, namely Jeddah, therefore not being very representative [127]. In certain cases, the surveys were repeated in a successive period in the same area. The mean acrylamide content of biscuits decreased in Belgium from 2002–2007 to 2008–2013 [129], but increased in the US from 2002–2006 to 2011–2015 [130].

**Table 4 foods-10-02751-t004:** Content of 5-hydroxymethyl furfural (HMF), furan, and acrylamide detected in biscuits in different national surveys.

Thermal Contaminant	Country	Year	Amount Detected	Reference
HMF	Slovakia	n.r.	0.34–34.99 mg/kg	[110]
	Spain	n.r.	3.1–182.5 mg/kg	[111]
Furan	Brazil	2009–2011	38.1–105.3 μg/kg	[112]
	UK	2012	4–165 μg/kg	[113]
	UK	2013	4–108 μg/kg	[113]
	Chile	n.r.	91 μg/kg	[114]
Acrylamide	Spain	2007–2019	20–2144 μg/kg	[121]
	Turkey	2004–2006	78–486 μg/kg	[122]
	Slovenia	2017–2018	20.5–3439 μg/kg	[123]
	Pakistan	n.r.	52–507 μg/kg	[124]
	Syria	2015	57–1433 μg/kg	[125]
	Colombia	n.r.	35–753 μg/kg	[126]
	Saudi Arabia	2015	90–182 μg/kg	[127]
	Czech Republic	n.r.	571–787 μg/kg	[128]
	Belgium	2002–2007	167 μg/kg (mean); 1514 μg/kg (maximum)	[129]
	Belgium	2008–2013	142 μg/kg (mean); 1113 μg/kg (maximum)	[129]
	US	2002–2006	119 μg/kg (mean) (n.d.–955 μg/kg)	[130]
	US	2011–2015	181 μg/kg (mean) (5–1796 μg/kg)	[130]

Maximum allowed level for biscuits = 350 μg/kg. HMF = 5-hydroxymethyl furfural; n.r. = not reported.

These findings have prompted extensive studies to find effective ways for reducing the level of acrylamide during food processing. In particular, acrylamide formation can be mitigated by (i) lowering the initial concentration of the acrylamide precursors in the raw materials; (ii) reformulating the food recipe; (iii) minimizing the thermal conditions of processing, which are a function of process temperature and time.

An effective mean for reducing acrylamide formation is to reduce the amount of free asparagine and reducing sugars of flours [131,132]. At this purpose, wheat genotypes with low levels of free asparagine were bred [133,134]. Otherwise, asparaginase added prior to processing can limit the acrylamide content by at least 70% [135,136]. More in detail, asparaginase lowered acrylamide formation by 23–75% depending on dough pH and incubation time. The optimum pH value for asparaginase action was in neutrality, and longer incubations (up to 60 min) were more effective. Additionally, flour milling intensity was taken into account, since it may enhance the content of reducing sugars. An effect of the refining degree of flour was also found, with lower contents of acrylamide in biscuits from refined flours than in the whole-grain ones [137]. Bran, in fact, was found to contain more free asparagine than refined flour [138]. Furthermore, rice contains less asparagine, therefore markedly lower acrylamide levels were found in rice biscuits (non-detected 204 μg/kg) than in wheat biscuits (155–661 μg/kg) [137].

Regarding the reformulation of products, different approaches were considered to inhibit acrylamide formation: lowering pH, exerting an antioxidant activity, or both. Organic acids, mainly citric [139], and aqueous rosemary extract were both effective in limiting the acrylamide content [140]. The addition of cations (calcium or magnesium salts) was also helpful to partially or completely eliminate the formation of the Schiff base [141]. In any case, it is important to replace ammonium bicarbonate (NH_4_HCO_3_) with other baking agents. Ammonium bicarbonate, indeed, promoted acrylamide formation [139] up to 6 times more than control biscuits without raising agents [142]. Alternative baking agents are combinations of sodium/potassium bicarbonate and organic acids. The replacement of inverted sugar syrup by a sucrose solution, instead, was an effective means for minimizing the presence of reducing sugars [139,143]. The addition of NaCl to biscuits had different results depending on the temperature. At 180 °C and 190 °C, the addition of NaCl reduced the acrylamide level, but not at 200 °C. The NaCl addition led to higher amounts of HMF at all the tested temperatures [144]. Adding cysteine (0.36 g/100 g) to the dough, alone or in combination with glycine (0.2 g/100 g), reduced the content of acrylamide and HMF in biscuits by 97.8% and 93.2%, respectively [145].

Biscuit-making industries, greatly interested in effective and simple ways for limiting acrylamide formation, are modifying their processing conditions to reduce the thermal load, while trying to keep good levels of sensorial satisfaction [146]. To minimize the thermal conditions of processing, researchers proposed several alternative baking technologies, such as vacuum baking [147], radio frequency (RF) heating [148], and microwave baking [149]. In particular, radio frequency post-drying of partially baked cookies (partially baked for 8 and 9 min and post-drying in a 27.12 MHz RF tunnel oven) was effective in lowering the acrylamide content compared to the control cookies (baked in a conventional oven at 205 °C for 11 min) [149]. Additionally, the adoption of an inhibitor atmosphere containing sulfur dioxide, which links the carbonyl groups of the reducing sugars, proved to be effective in reducing acrylamide formation during baking [150].

Other contaminants typically produced during dry heating, through the non-enzymatic reaction of sugars with proteins, are the advanced glycation end-products (AGEs), such as Nε-carboxymethyl-lysine (CML), Nε-1-carboxyethyl-lysine (CEL), and Nδ-(5-hydro-5-methyl-4-imidazolon-2-yl) ornithines (MG-H1) [151,152]. These compounds are generated in the late stages of Maillard reaction [153] and can be analyzed by ultra-performance liquid chromatography tandem mass spectrometry (UPLC-MS/MS) [154]. Crisp cereal products, such as biscuits, are among the top food groups contributing to AGEs intake [152], which may induce weight gain via insulin resistance and hypothalamic inflammation [155].

AGEs contribute also to Alzheimer’s disease, carcinogenesis, and degenerative bone disease [156]. The formation of AGEs is inhibited by antioxidants, including plant extracts and their bioactive compounds such as flavonoids, phenolic acids, tannins, curcumin, terpenes, silymarin, vitamin E, and vitamin C [157,158]. Recently, the incorporation of feruloylated oligosaccharides (FOs), which are novel food ingredients approved by the US Food and Drug Administration, has been proposed as a possible strategy to prevent excessive AGE formation in biscuits. FOs might inhibit the formation of AGEs in biscuits by exerting an antioxidant activity via the release of ferulic acid [159].

It is therefore possible to set industrial-level protocols for the reduction of newly formed contaminants of thermal origin, adapting the best strategy to the specific type of biscuit produced and to the consumption target. The reduction protocols, in fact, should be defined by reducing contaminants to safe levels but trying to minimize the alteration of sensory properties.

### 3.2. Thermal Increase of Pre-Existing Contaminants

Baking can raise the content of pre-existing contaminants, such as chloropropanols, PAH, and compounds linked to lipid oxidation (whose origin is discussed in Section 2.2). However, the increase of 3-MCPD, 2-MCPD, and MCPD in biscuits can be limited by lowering the thermal load during baking or limiting the chloride concentration (without using NaCl or reducing its amount) [62].

Additionally, the increase of PAH, generated during thermal steps of food preparation, such as baking, can be mitigated by lowering temperature and time [34].

Regarding lipid degradation, TGPs increased from 1.9 to 2.5 g/kg during baking, while the OxTGs raised from 5.8 to 6.0 g/kg [134]. Therefore, on the whole, the biscuit-making process has a relatively small impact on the oxidation level of raw materials [160]. The greatest effect on lipid oxidation is imputable to storage, discussed in Section 4.

### 3.3. Thermal Degradation of Pre-Existing Contaminants

The thermal effect of baking may have also a degradative/inactivating effect on some contaminants such as pesticides and mycotoxins possibly present in biscuits dough, decreasing their level. The specific fate of each pesticide during baking depends on its physicochemical properties. A recent study evaluated the effect of baking on 41 different pesticides. A reduction by about 24% was observed for most of them. Polar pesticides, such as carbendazim, and volatile compounds (chlorpyrifos-methyl, malathion, and pirimiphos-methyl) showed larger reduction rates, up to 33%. the greatest degradation of pesticides occurred in the first 6 min of baking. The highly volatile and polar pesticides, and those having a low degradation temperature, further decreased with prolonged baking time, reaching an almost complete elimination (95% decrease) [161].

Mycotoxins are characterized by a relatively high thermal and chemical stability, but baking may degrade them, depending on mycotoxin structure and processing conditions, in terms of temperature and duration. The concentration of DON and ZEN in biscuits was found to decrease with baking [162]. In detail, a 4–16% decrease of DON was observed during baking. Higher degradation was induced by increasing the content of NaHCO_3_ increased from 0.19% to 0.59%, and raising baking time and temperature from 7 min at 160 °C to 11 min at 200 °C [163]. The major degradation product was isoDON, which is considerably less toxic than DON, as shown by in vitro cell viability assays [163]. A 5% DON degradation in biscuit-making was reported by baking for 8 min at 180 °C, followed by 10 min at 100 °C [164]. After baking for 20 min at 180 °C the level of DON was found to decrease by 40%, with no significant improvement when a series of food additives were incorporated (such as ascorbic, citric, or sorbic acid, calcium propionate, lecithin) [165]. Regarding type-A trichothecenes, T-2 toxin showed a higher degradation rate than HT-2 toxin. During biscuit-making, up to 45% of T-2 toxin and 20% of HT-2 toxin were thermally degraded, with better results at higher baking time and temperature [166]. However, an increase of thermal load is not recommended to avoid the formation of thermal contaminants. Furthermore, it is worth noting that the decrease of mycotoxins does not necessarily always mean a decrease in toxicity, because the produced decomposition products could be as dangerous as the parent mycotoxins, and every case has to be verified. Therefore, achieving a high detoxification involves demonstrating that degradation products are effectively less toxic, while keeping good nutritional and organoleptic features at the same time. Besides, considering that OTA and TAF are characterized by high thermal stability, the most effective strategy is monitoring the raw materials to prevent the use of any contaminated batch.

## 4. Contaminants Deriving from Storage

During storage, two main phenomena can occur: (i) degrative reactions within the food matrix; (ii) release of chemicals from packaging. Some of biscuit constituents may undergo chemical degradations, such as lipid oxidation. Biscuit packaging therefore has to protect the product from water vapor, as well as from oxygen, to prevent moistening and lipid oxidation. Proper storage conditions, in a fresh and dry place, are also fundamental to prevent degradative reactions in biscuits.

The risks related to improper packaging could be also of microbiological nature. A bacterial increase was observed in biscuits packed in cellophane sacks (representing an insufficient barrier to oxygen and water vapor), stored at 20 °C and 62% relative humidity. In particular, the aerobic mesophilic bacteria increased from 2.1 × 10^2^ cfu/g to 7 × 10^5^ cfu/g in one month, reaching 1.3 × 10^6^ cfu/g in four months [167].

### 4.1. Oxidation-Related Contaminants

Lipid oxidation, slightly increased during baking, may further proceed during the shelf life of biscuits, raising the levels of OxTGs and TGPs. Biscuits, in fact, though not perceived as fatty foods, contain relevant levels of lipids, from 7.5% to 25% [71,94].

A study was carried out to evaluate the degradation of the lipid fraction of biscuits during storage. The study ascertained that storage contributed by 30% to the overall oxidative degradation of biscuits [73]. TGPs did not increase significantly, while OxTGs increased from 5.6–6.0 g/kg to 6.6–7.8 g/kg after 6 months of storage [73]. The oxidative stability was influenced at a higher extent by the initial fat content and storage time [168].

The most used packaging material for biscuits is orientated polypropylene (OPP). Acrylic-coating of OPP, alone or in combination with PVC/PVDC copolymer, provides a higher barrier to oxygen compared with plain OPP [169]. Metalized plastic films are used to limit the photooxidation [170]. Innovative materials containing poly-lactic acid (PLA) or OPP with ethylene vinyl acetate (EVA), were also proposed [171]. EVA is an additive, certified by FDA to come into food contact, used to make the plastic layer biodegradable. OPP with EVA was more effective in delaying lipid oxidation than PLL [171].

Besides using low oxygen permeable packaging materials, some reformulation strategies were proposed to limit lipid oxidation of biscuits. The replacement of NaCl with CaCl_2_ or KCl was found to decrease the oxidative degradation of lipids, due to prooxidant effect of NaCl [172]. The incorporation of natural ingredients bearing antioxidant properties or synthetic antioxidant, however, was the most frequently adopted strategy to inhibit oxidation and prevent nutritive losses [173]. For example, pomegranate peel [174] and mango peel, [175,176], rich in phenolic compounds, are proven to be effective in increasing the stability of biscuits. Plant extracts of chamomile and fennel conferred an antioxidant activity similarly to the synthetic antioxidant butylated hydroxyl anisole (BHA) [177]. Green tea extract was also proposed to limit the oxidation of biscuit lipids during storage [178]. Almond skins enriched biscuits of phenolics and raised the antioxidant activity [11]. Olive leaf extracts (OLE), rich of phenolics, exerted a strong antioxidant activity in baked dry snacks, slowing down the lipid oxidation. In detail, OLE induced a decrease of 27% in volatile compounds originated from oxidation and of 42% in TGPs compared to control [179,180]. The use of OLE could easily be transferred to biscuits, due to their similarity with baked dry snacks in terms of lipid content, moisture content, and production process. Besides conferring a relevant antioxidant activity, the use of OLE would allow also to reuse and add value to olive leaves, a waste of the olive oil industry. This approach is increasingly important in a circular economy perspective applied to the food industry.

### 4.2. Contaminants Released from Packaging

Regarding the possible release of chemicals from packaging, MOH contamination takes place if recycled paper and board are used, without an internal bag to contain biscuits. These packaging materials, in fact, could bring some chemicals from the printing inks and adhesives into direct food contact. Printing ink used for decorating boxes or paper bags may contain both MOSH and MOAH [147].

Internal bags made from polypropylene, acrylate-coated polypropylene, polyethylene terephthalate (PET), alone or coupled to aluminum layer avoid or at least reduce the contamination from paperboard box. Internal bags of paper and/or polyethylene (PE) do not reduce the migration of MOSH compared to control packs without an internal bag [181]. Compared to other cereal-based food products, the highest migration from paperboard to food was observed in biscuits due to their typically high fat content. Lower migration occurred to less fatty foods such as egg-based pasta, or wheat and rice flour [182].

Other solutions to mitigate this issue involve using virgin board or, alternatively, using recycled paperboard incorporating an activated carbon layer, which reduces MOH migration [183,184].

## 5. Conclusions and Future Perspectives

Biscuits, like most foods, are a possible source of a wide range of contaminants, which may cause potential health risks. The most critical step of the production process is represented by the selection of raw materials. The successive processing steps must be monitored as well, because if they cannot improve the initial safety condition, they could worsen it.

Fast methods, such as the visible/near infrared hyperspectral imaging [185] or tailored optical sensors based on fast Fourier transform (FFT) spectrum analysis [186], are increasingly needed to automatically check raw materials (though they should be certified to meet current legal restrictions) as they arrive at industries. Ideally, similar automatic control systems should also be available online, to monitor the contaminants formed during processing.

To limit the contaminations, several mitigation strategies have been set up, and many others are under study, also taking advantage of innovative technologies. To reduce mycotoxins in the raw materials, cereal grains could be treated by UV, ozone, pulsed light, cold atmospheric pressure plasma, or can be added of adsorption agents. During refining of vegetable oils used in biscuit-making, PAH could be removed by adding activated charcoal, whereas MCPD and GE could be reduced by adopting milder thermal treatments. Furthermore, legal maximum limits should be established for these contaminants also in biscuits, and not only in the oils.

The most effective mitigation approaches during the production process, instead, involve product reformulation, and the use of alternative baking technologies (vacuum baking, radio frequency heating) to minimize the thermal load which causes the formation of contaminants such as acrylamide. These approaches could allow establishing effective reduction protocols at industrial level, which should be always set up by trying to keep the sensory properties of finished products acceptable.

The use of low oxygen permeable packaging materials (avoiding the direct contact with recycled ones), and product reformulation by incorporating antioxidants are, instead, the most effective strategies for limiting the raise of oxidation-related contaminations during biscuit storage.

Some process modifications—coupled with continuous monitoring of raw materials, intermediates, finished products, and processing conditions—are therefore essential to meet current regulatory restrictions, achieving the aim of banning dietary contaminants and coping with chronic diseases.

## Figures and Tables

**Figure 1 foods-10-02751-f001:**
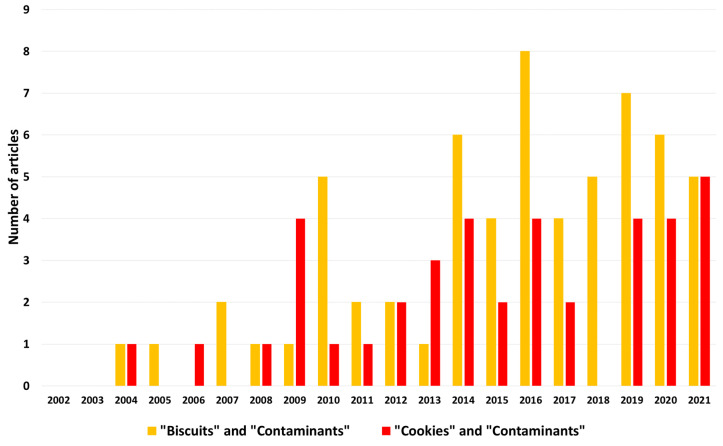
Number of articles published in the last 20 years (2002–2021) containing the words “biscuits” or “cookies” and “contaminants” in the title, abstract and keywords. Source = Scopus database (Elsevier).

**Figure 2 foods-10-02751-f002:**
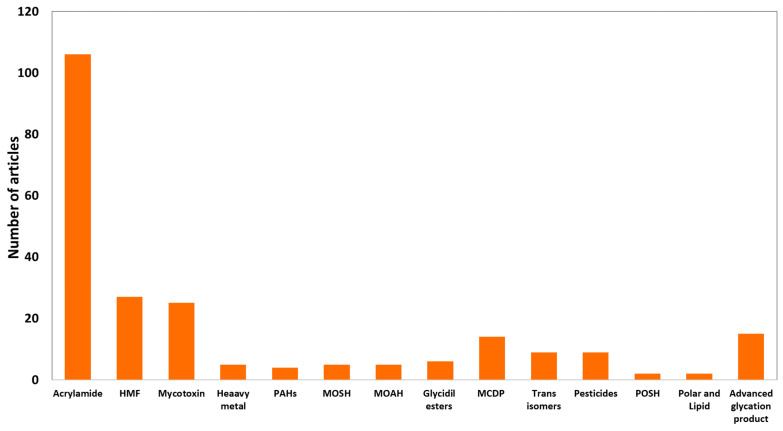
Number of articles published in the last decade (2012–2021) containing in the title, abstract, and keywords the words “biscuits” and the name of each contaminant. Source = Scopus database (Elsevier). HMF = 5-hydroxymethyl furfural; PAHs = polycyclic aromatic hydrocarbons; MOSH = mineral oil saturated hydrocarbons; MOAH = mineral oil aromatic hydrocarbons; MCPD = monochloropropanediol esters; POSH = polyolefin oligomeric saturated hydrocarbons.

**Figure 3 foods-10-02751-f003:**
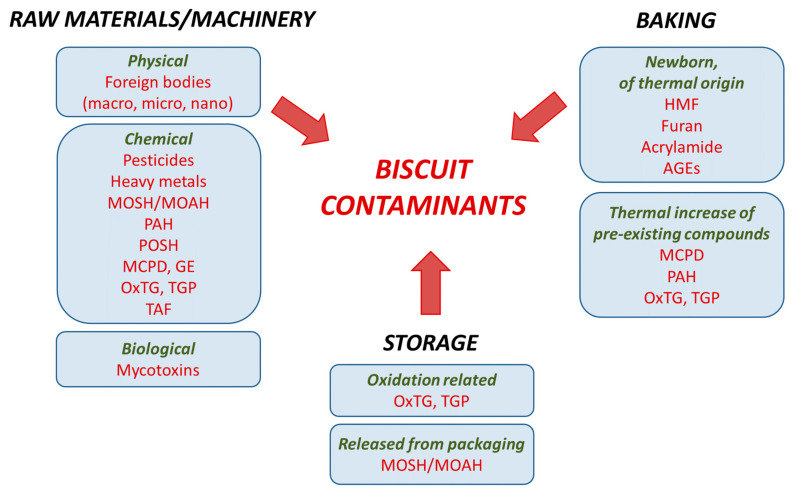
Possible biscuit contaminants from raw materials/machinery, baking process, and storage. MOSH = mineral oil saturated hydrocarbons; MOAH = mineral oil aromatic hydrocarbons; PAHs = polycyclic aromatic hydrocarbons; POSH = polyolefin oligomeric saturated hydrocarbon; MCPD = Monochloropropanediol esters; GE = glycidyl esters; OxTG = oxidized triglycerides; TGP = triglyceride oligopolymers; TFA = *trans* fatty acids; HMF = 5-hydroxymethyl furfural; AGEs = advanced glycation end-products.

**Table 1 foods-10-02751-t001:** Content of lead and cadmium detected in biscuits in different national surveys.

Element	Country	Amount Detected (mg/kg)	Reference
Lead	Nigeria	0.10–0.16 ^1^	[32]
	Nigeria	2.8	[33,34]
	India	0.13	[35]
	Egypt	0.12	[36]
Cadmium	Nigeria	0.02–0.03 ^2^	[32]
	Nigeria	0.35	[33,34]
	India	n.d.	[35]
	Greece	0.01	[36]
	Egypt	0.01–0.12	[37]

^1^ Maximum allowed level for cereal grains = 0.20 mg/kg; ^2^ Maximum allowed level for cereal grains = 0.10 mg/kg; n.d. = not detected.

**Table 2 foods-10-02751-t002:** Legislative limits for mycotoxins in cereals and cereal-based foodstuffs according to the EC Reg. 1881/2006 [38] and the Commission Recommendation 2013/165/EU [80].

Mycotoxin	Food Product	Maximum Level (μg/kg)
B_1_ Aflatoxin	All cereals and all processed cereal products, with the exemption of maize	2.0
	Maize	5.0
	Processed cereal-based foods for infants and young children	0.10
Sum of B_1_, B_2_, G_1_ and G_2_ Aflatoxins	All cereals and all processed cereal products, with the exemption of maize	4.0
	Maize	10.0
Ochratoxin A	Unprocessed cereals	5.0
	All processed cereal products	3.0
	Processed cereal-based foods for infants and young children	0.50
Deoxynivalenol	Unprocessed cereals other than durum wheat, oats, and maize	1250
	Unprocessed durum wheat and oats	1750
	Unprocessed maize	1750
	Cereal flour	750
	Bread, pastries, biscuits, cereal snacks, and breakfast cereals	500
	Processed cereal-based foods for infants and young children	200
Zearalenone	Unprocessed cereals other than maize	100
	Unprocessed maize	200
	Cereal flour	75
	Bread, pastries, biscuits, cereal snacks, and breakfast cereals	50
	Processed cereal-based foods for infants and young children	20
Sum of T-2 and HT-2	Unprocessed barley and maize	200
	Unprocessed wheat and rye	100
	Bread, pastries, biscuits, cereal snacks and breakfast cereals, pasta	25
	Cereal-based foods for infants and young children	15

**Table 3 foods-10-02751-t003:** Microbiological specifications established by the World Food Program for high energy biscuits [96].

Microorganism	Maximum Level
Aerobic mesophilic bacteria	10^4^ cfu/g
Coliforms	10 cfu/g
*Escherichia coli*	0 cfu/10 g
*Salmonella*	0 cfu/25 g
*Staphylococcus aureus*	10 cfu/g
*Bacillus cereus*	10 cfu/g
Yeasts and molds	10^2^ cfu/g

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
