# Peer review of "Biscuit Contaminants, Their Sources and Mitigation Strategies: A Review"

_foods, 2021, doi:10.3390/foods10112751_

Round 1

Reviewer 1 Report

Dear authors, 

the present work was very interesting and well done. 

I would like to suggest only one modification.
The guidelines indicate, as a maximum number of 10 keywords.
I suggest removing "biscuit", because this word is already present in the title

Author Response

Reviewer 1.

Dear authors, 

the present work was very interesting and well done. 

Response. Thanks a lot for appreciating our work.

I would like to suggest only one modification.
The guidelines indicate, as a maximum number of 10 keywords.
I suggest removing "biscuit", because this word is already present in the title

Response. Thanks for noting. The word “biscuit” has been deleted.

Reviewer 2 Report

Comments:

In this manuscript, the author reviews the current state of the scientific literature on the possible contaminants of biscuits. It also describes physical, chemical, and biological hazards, and critically analyzes solutions to reduce such contaminates. However, I think that this research is too simple and not in-depth. The author should add more self-discussions in the review to show the meaning and depth of the manuscript. And many sentences in the manuscript are very confusing and English should be revised. The manuscript needs major revisions.

Other comments:

  1. Spacing, punctuation marks, grammar, and tenseerrors should be reviewed thoroughly.
  2. Rewrite the very long sentences throughout the entire manuscript.
  3. The whole manuscript must be evaluated by a native speaker of English.
  4. The discussion is very superficial, thus a better discussion on the results obtained is crucial with a comparison of them. It is necessaryto illustrate the novelty of this research. Try to include the existing research limitations.
  5. I suggest the authors discuss the most significant results more depth, increase more of your own discussion, and explain your ideas in an orderly manner. It can make the article more complete and sufficientlyunderstood

Author Response

Reviewer 2.

In this manuscript, the author reviews the current state of the scientific literature on the possible contaminants of biscuits. It also describes physical, chemical, and biological hazards, and critically analyzes solutions to reduce such contaminates. However, I think that this research is too simple and not in-depth. The author should add more self-discussions in the review to show the meaning and depth of the manuscript. And many sentences in the manuscript are very confusing and English should be revised. The manuscript needs major revisions.

Response: More self-discussions have been added. English has been revised by a native English speaker.

Other comments:

    Spacing, punctuation marks, grammar, and tense errors should be reviewed thoroughly.

Response: Spacing, punctuation marks, grammar, and tense errors have been reviewed thoroughly and corrected.

    Rewrite the very long sentences throughout the entire manuscript.

Response: Very long sentences have been rewritten to shorten them.

    The whole manuscript must be evaluated by a native speaker of English.

Response: The whole manuscript has been revised by a native English speaker.

    The discussion is very superficial, thus a better discussion on the results obtained is crucial with a comparison of them. It is necessary to illustrate the novelty of this research. Try to include the existing research limitations.

Response: The novelty of the research has been highlighted at the end of the Introduction (lines 80-85). Discussion has been deepened (lines 311-320, 332-335, 537-541, 589-593, 638-643, 662-291). Some existing research limitations have been commented (lines 217-220; 224-226; 233-241).

    I suggest the authors discuss the most significant results more depth, increase more of your own discussion, and explain your ideas in an orderly manner. It can make the article more complete and sufficiently understood

Response: Discussion has been deepened (lines 311-320, 332-335, 537-541, 589-593, 638-643, 662-291).

Reviewer 3 Report

It is a very interesting and well-written review, which fits well to the scope of the Foods Journal.

Several comments are below:

Make sure you always use dot with digitals, because I have seen both in the manuscript f.ex. line 215 “2,0 and 10,0” also line 267

In my opinion, under each Figure, the meaning of all used abbreviations should be explained.

 Maybe a separate list of abbreviations used in the manuscript would help, because there are a lot used in the paper and sometimes it might be difficult to quickly understand the meaning.

Line 333, a dot before “However” is missing

Line 487-488: mentioning different kinds of pasta and the migration to them is not within the scope of the review. I would suggest removing this part or replacing it with something more related to biscuits.

I think the "Conclusions" could be still improved. Try to sum up everything or point out the most important information of the review.

Reviewer 4 Report

Determination of physical, chemical, and biological contaminants in biscuit products and development and critical analysis of the efficiency of methods for decontamination has high relevance in the field of food science and technology. Authors of the manuscript foods-1452001 focused on the origin of contaminants, possible solutions for decontamination, effects of the technological parameters and machinery, and the role of appropriate packaging materials. Therefore, the topic of the manuscript is relevant and MS can provide interesting information for the readers.

The manuscript is generally well written with a logic structure. The relevancies and objectives of the study are well defined in the Introduction section. Section 2 provides a good summary on the different contaminants. Section 3 gives a good summary about the effects of baking technology on contaminants. Section 4 deals with the changes arisen during the storage.  Establishments are based on relevant references.

Comments, suggestions:

In my opinion, section 2.3 ’Biological contaminants’ is a little superficial. I suggest the authors to give more detailed information on microbial contaminants and mycotoxin problems (and/or give the characteristics data, legislative limits in a table).

I suggest the authors to give more information related to applicability of antioxidants and antimicrobial agents for reduced oxidation etc. (minimize the negative effects of chemical, microbiological and sensory properties).

I recommend the authors to give DOIs for all every references.

Author Response

Reviewer 3

Determination of physical, chemical, and biological contaminants in biscuit products and development and critical analysis of the efficiency of methods for decontamination has high relevance in the field of food science and technology. Authors of the manuscript foods-1452001 focused on the origin of contaminants, possible solutions for decontamination, effects of the technological parameters and machinery, and the role of appropriate packaging materials. Therefore, the topic of the manuscript is relevant and MS can provide interesting information for the readers.

The manuscript is generally well written with a logic structure. The relevancies and objectives of the study are well defined in the Introduction section. Section 2 provides a good summary on the different contaminants. Section 3 gives a good summary about the effects of baking technology on contaminants. Section 4 deals with the changes arisen during the storage.  Establishments are based on relevant references.

Response. Thanks a lot for appreciating our work.

Comments, suggestions:

In my opinion, section 2.3 ’Biological contaminants’ is a little superficial. I suggest the authors to give more detailed information on microbial contaminants and mycotoxin problems (and/or give the characteristics data, legislative limits in a table).

Response. Thanks for suggestion. Section 2.3 has been extensively rewritten by adding more detailed information on microbial contaminants (lines 376-412) and mycotoxin problems (line 332-375). Mitigation strategies of mycotoxins in cereals have been also added, which were not properly reviewed in the previous version of the manuscript.

A Table illustrating the legislative limits for each mycotoxin and for both unprocessed and processed cereal-based food, including biscuits, has been added (Table 2). A Table with the maximum limits for microbiological contaminants has been also added (Table 3).

I suggest the authors to give more information related to applicability of antioxidants and antimicrobial agents for reduced oxidation etc. (minimize the negative effects of chemical, microbiological and sensory properties).

Response: More information related to the use of antioxidants have been given (lines 627-643). More information related to the use of antimicrobial agents have been given (lines 396-400).

I recommend the authors to give DOIs for all every references.

Response: The doi numbers have been added.

Round 2

Reviewer 2 Report

The authors addressed all of my comments. The improved version of the manuscript is suitable for publication.